# Genome-Wide Identification of the *CAT* Genes and Molecular Characterization of Their Transcriptional Responses to Various Nutrient Stresses in Allotetraploid Rapeseed

**DOI:** 10.3390/ijms252312658

**Published:** 2024-11-25

**Authors:** Xiao-Qian Du, Si-Si Sun, Ting Zhou, Lu Zhang, Ying-Na Feng, Kun-Long Zhang, Ying-Peng Hua

**Affiliations:** School of Agricultural Sciences, Zhengzhou University, Zhengzhou 450001, China; duxiaoqian2022@163.com (X.-Q.D.); sunsisi13333910080@163.com (S.-S.S.); zhoutt@zzu.edu.cn (T.Z.); zhanglu2102738@163.com (L.Z.); yingnafeng@zzu.edu.cn (Y.-N.F.)

**Keywords:** *Brassica napus*, cationic amino acid transporters, genome-wide analysis, expression pattern

## Abstract

*Brassica napus* is an important oil crop in China and has a great demand for nitrogen nutrients. Cationic amino acid transporters (CAT) play a key role in amino acid absorption and transport in plants. However, the *CATs* family has not been reported in *B. napus* so far. In this study, genome-wide analysis identified 22 *CAT* members in the *B. napus* genome. Based on phylogenetic and synteny analysis, *BnaCATs* were classified into four groups (Group I–Group IV). The members in the same subgroups showed similar physiochemical characteristics and intron/exon and motif patterns. By evaluating *cis*-elements in the promoter regions, we identified some *cis*-elements related to hormones, stress and plant development. Darwin’s evolutionary analysis indicated that *BnaCATs* might have experienced strong purifying selection pressure. The *BnaCAT* family may have undergone gene expansion; the chromosomal location of *BnaCATs* indicated that whole-genome replication or segmental replication may play a major driving role. Differential expression patterns of *BnaCATs* under nitrate limitation, phosphate shortage, potassium shortage, cadmium toxicity, ammonium excess and salt stress conditions indicated that they were responsive to different nutrient stresses. In summary, these findings provide a comprehensive survey of the *BnaCAT* family and lay a foundation for the further functional analysis of family members.

## 1. Background

Amino acids are not only a source of nitrogen (N) nutrients that can be directly absorbed by plants but are also the major transport form of organic n in plants [1]. Amino acids are distributed within the plant through both xylem and phloem to supply organs that are net importers, such as seeds or tubers [2]. The two major amino acid transporter families in plants can be classified as *ATF* (amino acid transporter family) and *APC* (amino acid polyamine choline transporters) [3]. Among them, cationic amino acid transporters (CAT) belong to the *APC* family and play a key role in amino acid transport and N metabolism in plants [4]. In *Arabidopsis thaliana*, the *CAT* family contains nine protein members (*AtCAT1*–*9*) that are widely expressed in different tissues including roots, stems, leaves, flowers and fruits of *A. thaliana* [5,6,7,8,9,10]. *AtCAT1* (cationic amino acid transporter 1) plays a role in nitrogen distribution and balance [5]. *AtCAT2* is a critical target of leaf amino acid concentrations, and manipulation of this tonoplast transporter can significantly alter total tissue amino acid concentrations [6]. *AtCAT5* functions as a high-affinity basic amino acid transporter at the plasma membrane [7]. *AtCAT3* and *AtCAT4* play roles in intracellular compartmentalization of amino acids, intercellular transport across plasmodesmata and loading/unloading of vascular tissue, respectively [8]. *AtCAT6* and *AtCAT8* are involved in the delivery of amino acids to library tissues [9]. *AtCAT9* plays an important role in amino acid homeostasis [10]. In wheat, ten identified CAT proteins may be related to the response to various stresses, are cytoplasm localized and may function as antioxidant enzymes [11]. Four proteins have been identified in soybean that play important roles in plant defense, development, and senescence [12]. Four candidate *CsCATs* were identified in cucumber. Based on the expression pattern comparison, *CsCATs* exhibited an expression pattern similar to *Arabidopsis* counterparts [13]. In addition, the ectopic expression of maize *CAT2* (*ZmCAT2*) in tobacco can induce *CATs* activity, improving pathogen resistance [14]. The allotetraploid *Brassica napus* (A_n_A_n_C_n_C_n_, 2n = 4x = 38) is the second most important oilseed crop in the world and originated from the spontaneous hybridization of the diploid *B. rapa* (A_r_A_r_, 2n = 2x = 20) and *B. oleracea* (C_o_C_o_, 2n = 2x = 18) [15,16,17]. Rapeseed is one of the most important oil crops in China, and its area and production rank first in the world. It can provide about 5 million tons of edible oil for China annually. Increasing the yield per unit area is an important way to increase output value per unit area of rapeseed [18]; its yield and quality are greatly affected by the N element. Due to the low efficiency of N use in rapeseed, a large amount of N fertilizers should be invested to ensure rapeseed yield [7]. Therefore, improving the N remobilization efficiency in oilseed rape is important for increasing N use efficiency through molecular modulation of amino acids transporters, especially *CATs*.

However, there are few systematic analyses of *CATs* in *B. napus*. Therefore, it is very important to analyze the nutritional physiological and biological characteristics of *CAT* members in *B. napus*. In this study, we aimed to (i) identify the genome-wide *CATs* in *B. napus*, (ii) characterize the genomic properties and transcriptional responses of *CAT* members to N stresses, including nitrate limitation and ammonium toxicity, and (iii) investigate the transcriptional responses of *CATs* to other nutrient stresses, including phosphate limitation, boron deficiency, cadmium toxicity and salt stress. Bioinformatics and molecular biology methods were used to identify compare and analyze the expression of the *B. napus CATs* (named as *BnaCAT*). Through the statistics of the transmembrane region of BnaCAT proteins, active site prediction, phylogenetic analysis, protein interaction and expression pattern exploration, such data can provide a partial reference for related studies of amino acid transport and nitrogen nutrient metabolism in rapeseed.

## 2. Results

### 2.1. Identification of BnaCAT Family Members and Construction of Phylogenetic Tree in B. napus

According to the protein sequences of the *AtCATs* family in *A. thaliana*, 22 *BnaCAT* members have been identified in the *B. napus* genome. According to the sequence on the *B. napus* chromosome, the 22 *BnaCATs* were renamed. Furthermore, using the amino acid sequences of *Arabidopsis* family members as query conditions, we screened and identified the homolog of *B. oleracea* and *B. rapa* in the BRAD database through the PFAM domain. As shown in Table 1, *CATs* had nine members (*CAT1*–*CAT9*) in the *A. thaliana* model and each *CAT* member had a single copy. A total of 13, 13, and 22 *CAT* homologues were identified in *B. rapa*, *B. oleracea,* and *B. napus*, respectively. The results showed that the number of *CATs* in *B. napus* was similar to the sum of *CATs* in *B. rapa* and *B. oleracea*. This suggests that most *CATs* were conserved during spontaneous hybridization between *B. rapa* and *B. oleracea* to from the allotetraploid *B. napus*. However, we found that *CAT7* was lost in *B. napus*. The changes in the number of *BanCATs* may indicate their critical differential roles in the resistance of *B. napus* to N stress. To explore the phylogenetic relationship between 22 members of the *CATs* family, an unrooted phylogenetic tree was constructed using 10.2.2 (Figure 1). We performed phylogenetic analysis of CAT proteins in *A. thaliana* and *B. napus*. The phylogenetic tree could be divided into four major clades, which could be further subdivided into nine smaller categories, and each *BnaCAT* member was closely clustered with the corresponding homologs in *A. thaliana* (Figure 1). There were no homologous genes of *AtCAT7* in *B. napus*, suggesting that the homologs in *B. napus* were lost during evolutionary history. In conclusion, these results showed that the *CAT* family in *B. napus* underwent specific evolutionary events after the divergence of *A. thaliana*.

### 2.2. Molecular Characterization of BnaCATs

In order to gain insight into the molecular characteristics of the BnaCAT proteins, we calculated the physicochemical parameters of each BnaCAT protein using ExPASy. The results demonstrated that the majority of proteins within the same *CAT* subfamily exhibited comparable physicochemical parameters (Table 2). In conclusion, the coding sequence (CDS) lengths of *BnaCATs* exhibited considerable variation, ranging from 1590 bp (*BnaA3.CAT6* and *BnaC3.CAT6*) to 1917 bp (*BnaA9.CAT2a*). This variation was reflected in the deduced amino acid (AA) number, which ranged from 529 (*BnaC3.CAT6*, *BnaC3.CAT6*) to 638 (*BnaA9.CAT2a*) (Table 2). The computed molecular weights of *BnaCATs* exhibited a range from 56.9 KD (*BnaA3.CAT6* and *BnaC3.CAT6*) to 68.0 KD (*BnaCn.CAT3*) (Appendix A). Theoretical isoelectric points (pIs) of *BnaCATs* exhibited a range from 5.56 (*BnaAn.CAT4*) to 9.02 (*BnaC7.CAT6*), with some values exceeding 7.0 (Appendix A). The GRAVY index reflects the hydrophilic and hydrophobic nature of the protein physicochemical properties. The results showed that the GRAVY values of the *BnaCATs* ranged from 0.512 (*BnaC7.CAT1*) to 0.795 (*BnaC3.CAT6*) (Appendix A). Therefore, it can be assumed that all the CAT proteins in *Brassica napus* are hydrophobic. The majority of *BnaCATs* exhibited instability 40.0 (Appendix A), indicating that the majority of *BnaCATs* demonstrated robust protein stability, with the expectation of *BnaAn.CAT9* (41.71) to exhibit an instability index above >40.0. One study revealed the subcellular localization of 9 *AtCATs*, while the online WoLF PSORT was employed to predict the subcellular localization of 22 *BnaCATs* (Appendix A). The finding indicated that the majority of them were localized in the plasma membrane, suggesting that they might be involved in the trans-membrane transport of AAs. In detail, *BnaAn.CAT9* was mainly distributed in the plasma membrane and endoplasmic reticulum, suggesting that it might play an important role in amino acid homeostasis. We utilized the TMHMM tool to characterize the transmembrane structures of *CATs* in *A. thaliana* and *B. napus*, and found that *AtCATs* and *BnaCATs* had twelve to fifteen membrane-spanning regions (Appendix A). In detail, *AtCAT2*/*BnaCAT2s*, *AtCAT3*/*BnaCAT3s*, *AtCAT4*/*BnaCAT4s* and *AtCAT5*/*BnaCAT5s* had fourteen trans-membrane regions. *AtCAT8*/*BnaCAT8s* had thirteen trans-membrane regions, and the other three subgroup members had different membrane-spanning regions between *Arabidopsis* and *B. napus.* The NetPhos tool was employed to identify phosphorylation sites in *BnaCATs*. The results indicated that serine is the most prevalent site for phosphorylation (Appendix A). In line with *AtCATs* lacking signal peptides, *BnaCATs* were also found to lack any signal peptides (Appendix A).

### 2.3. Identification of Evolutionary Selection Pressure on BnaCATs

To explore the selective pressure on *BnaCATs*, the non-synonymous/synonymous mutation ratio (Ka/Ks) was calculated; Ka/Ks > 1.0 indicates positive selection, Ka/Ks = 1.0 indicates neutral selection, and Ka/Ks < 1.0 indicates purifying selection (Table 2).

The Ka values of *BnaCATs* exhibited a wide range, from 0.0362 (*BnaC3.CAT6*) to 0.429 (*BnaC7.CAT6*), with an average of 0.0709. Similarly, the Ks values of *BnaCATs* demonstrated considerable variation from 0.2769 (*BnaAn.CAT9*) to 2.1475 (*BnaC7.CAT6*), with an average of 0.5433. Furthermore, it was observed that all Ka/Ks values of *BnaCATs* were less than 1.0 (Table 2). Consequently, it was postulated that the *BnaCATs* may have been subjected to a particularly intense negative selection pressure in order to maintain their functionality. The Ks values of duplicated homologs among gene families are typically regarded as molecular clocks, with the assumption that they remain unaltered over time. The divergence between the model *Arabidopsis* and its derived *Brassica* species is estimated to have occurred approximately 12–20 million years ago (Mya) [19,20]. The results indicated that the majority of *BnaCATs* diverged from *AtCATs* approximately 11.0–20.0 Mya (Figure 1), suggesting that the divergence of the *Brassica* species may have occurred concurrently with the divergence of the *CATs*.

### 2.4. Chromosomal Distribution and Syntenic Analysis of BnaCATs

Gene expansion is a phenomenon observed during the evolution of species [21]. According to chromosomal annotation information of *B. napus*, *CATs* identified were mapped on chromosomes (Figure 2). Twenty-two *BnaCATs* were distributed unevenly in the *B. napus* genome, with each chromosome containing one to three genes (Figure 2). Chromosome Ann random had the highest number of *BnaCATs*, with three *BnaCATs*. Furthermore, the distribution of A and C chromosomes in *B. napus* was also unequal, with 12 in the A genome and 10 in the C genome (Figure 2).

Gene family expansion occurs primarily through four pathways: tandem replication, fragment replication, whole genome replication (polyploidy) and replication transposition [4]. Gene duplication plays a pivotal role in plant evolution. Comparative genomics has revealed that the *Arabidopsis* genome can be divided into 24 ancestral cruciferous blocks, labeled A–X [22]. The results demonstrated that *CATs* family members in *Arabidopsis* and their corresponding homologues in *B. napus* are located in the same chromosomal segment (Table 2). The inter-chromosomal relationship of *BnaCATs* exhibited 18 pairs of segmental duplications. These findings indicated that gene replication played a pivotal role in the amplification of *CATs* in the *B. napus* genome, with whole genome replication or segmental replication may serve as a primary driving force.

The evolution of the *CATs* gene family in the genus *Brassica* was investigated by analyzing the homologous relationships among *B. napus*, *A. thaliana*, *B. rapa*, and *B. oleracea*. A collinear analysis revealed that a considerable number of homologous *CATs* were present in *B. napus*, *A. thaliana*, *B. rapa*, and *B. oleracea*. A total of twelve pairs of genes were collinear in *B. napus* and *A. thaliana* (Figure 3), and 11 *CATs* in *B. napus* had homologous genes in *A. thaliana* (Figure 3). Furthermore, 23 pairs of genes in *B. napus* and *B. rapa* demonstrated collinearity (Figure 3). Of the 16 *B. napus CATs* with homologous genes in *B. rapa* (Figure 3), there were 23 pairs of genes in *B. napus* and *B. oleracea*, and 72.7% (16) of *CATs* in *B. napus* had homologs in *B. oleracea* (Figure 3). These findings suggest that the majority of *CATs* remained intact throughout the formation and evolution of *B. napus*.

### 2.5. Conserved Motifs, Gene Structure Analysis of BnaCATs

To further clarify the potential functions of *CATs* in *B. napus*, MEME was used to identify 15 conserved motifs. We found that the amino acid sequences of the motifs 1, 2, 3, 5, 6, 7, 8, 12, and 14 had the highest identity among all the *BnaCATs* (Figure 4), and thus might be used as indicators of the *CAT* family members. The genetic classification revelated that the CAT proteins exhibited similarities among the four groups, while there were also differences among the groups (Figure 4). For instance, motif 15 was specific to group I and motif 9 was specific to the group IV (Figure 4). This indicated that the *CAT* sequence is evolutionarily conserved but differentiated.

To evaluate the sequence diversity of *BnaCATs*, the exon–intron structures of each *BnaCAT* were detected. In detail, the number of introns and exons varies among each group of *BnaCATs* (Figure 4). It was observed that similar structures were typically found within the same group (Figure 4). The number of introns in Group I is 13, with the exception of *BnaA4.CAT3* (Figure 4). The *BnaAn.CAT9* sequence exhibited six introns (Figure 4). The Group III genes exhibited two or four introns (Figure 4). The number of introns present in the Group IV genes exhibited a range from zero to two (Figure 4). These results indicated the clusters of *BnaCATs* had a similar intron/exon pattern. Studies on the conserved motif composition, gene structure and phylogenetic relationship have demonstrated that BnaCAT proteins have very conserved amino acid residues, and members within the group may have similar functions.

### 2.6. Cis-Element Analysis of the Promoter Regions of the BnaCATs

*Cis*-acting elements play a key role in the regulation of gene expression. To investigate the function and regulatory patterns, a 2000 bp sequence of these genes was submitted to the PlantCare database. The *cis*-elements of *BnaCATs* were primarily classified into three categories: plant growth and development, stress-responsive elements and phytohormone responsive elements. The first category of elements primarily encompasses meristem expression (CAT-box), zein metabolism regulation (O_2_-site), endosperm expression (GCN4-motif) and flavonoid biosynthetic gene regulation (MBSI) (Figure 5). In the second category, which pertains to stress-responsive elements, the following elements were identified: wound-responsive (WUN motif), anaerobic induction (ARE), low-temperature-responsive (LTR) and MYB-binding sites involved in drought inducibility (MBS), and stress responsiveness (TC-rich repeats) (Figure 5). In the second category (phytohormone responsive), the elements included gibberellin responsiveness (P-box) and methyl methyl jasmine-responsive (CGTCA-motif), auxin-responsive (TGA-element) and abscisic acid-responsive (ABRE) (Figure 5) elements. Among these *cis*-elements, ABRE-, ARE-, and CGTCA-motif elements were particularly noteworthy, as they were involved in abscisic acid responsiveness, anaerobic induction and MeJA responsiveness (Figure 5). These results indicate that *BnaCATs* may be induced or repressed by abiotic stresses, subsequently participating in plant stress resistance. It is noteworthy that each *BnaCAT* exhibited a distinct array of *cis*-elements; this suggested that under varying growth and developmental stages, environmental conditions and other factors, *BnaCATs* may function independently or in concert to ensure optimal plant growth and development.

### 2.7. Protein–Protein Interaction Analysis of BnaCATs

To further identify the protein(s) potentially interacting with the *CAT* family members, we constructed a protein interaction network of *CATs* using the STRING database. As illustrated in Figure 6, the proteins closely related to CAT proteins in *Arabidopsis thaliana* are primarily polyamine absorption transporters (PUT, polyamine uptake transporter) and certain amino acid permeases (AAPs). AAPs play roles in the transport of a wide range of amino acids and the regulation of physiological processes in plants [21]. The secondary structures of *CATs* were predicted using the SOPMA [4]. The three-dimensional structure of BnaCAT proteins was predicted using the Phyre2 software V2.0 (https://www.sbg.bio.ic.ac.uk/, accessed on 10 September 2024) (Appendix A). The secondary structures of the eight BnaCAT proteins were found to be composed of four structural elements: α-helix, extended chain, β-fold, and random curl (Appendix A). The percentages of α secondary structures ranged from 39.78% (*BnaA4.CAT3*) to 55.05% (*BnaC7.CAT1*), with an average of 48.39% (Appendix A). The random coil ratios of *BnaCATs* exhibited considerable variation, ranging from 27.54% (*BnaAn.CAT5*, *BnaA4.CAT5*) to 39.87% (*BnaA9.CAT2a*), with an average of 32.26% (Appendix A). The proportion of β-fold in *BnaCATs* ranged from 3.36% (*BnaA3.CAT1*) to 5.44% (*BnaAn.CAT9*), with an average of 4.17% (Appendix A). This suggests that the α-helix constitutes a significant component of the *BnaCAT* secondary structure, with the random coil and β-fold elements being the least abundant. Eight BnaCAT proteins exhibited a high proportion of random coil structure (Appendix A), which is often affected by the side chain to form the active site of the protein [22].

### 2.8. Expression Profiles of BnaCATs in Response to Diverse Nutrient Stresses

To identify the expression patterns of *BnaCATs*, we initially investigated the tissue-specific expression patterns of *BnaCATs* in various tissues through the BnIR. The results demonstrated that *BnaCATs* exhibited distinct expression patterns. *BnaA9.CAT8*, *BnaC8.CAT8* and *BnaAn.CAT9* were found to be constitutively expressed in multiple tissues, whereas others displayed preferential expression in specific tissues (Appendix A). For example, *BnaC7.CAT1*, *BnaA3.CAT1*, *BnaA9.CAT2a*, *BnaCn.CAT2*, *BnaCn.CAT3* and *BnaA5.CAT4* were found to be predominantly expressed in senescent leaves and dried seeds (Appendix A). The *BnaA8.CAT1*, *BnaC3.CAT1*, *BnaAn.CAT4*, *BnaC1.CAT4, BnaA9.CAT2b*, *BnaA4.CAT3*, *BnaC9.CAT2* and *BnaC3.CAT6* genes were found to be highly expressed in buds or flowers, indicating a potential involvement in seed development (Appendix A). *BnaC5.CAT4* was predominantly expressed in buds and dried seeds (Appendix A). The preferential expression of *BnaA4.CAT5* in the lower stem indicated its participation in long-distance translocation of AA (Appendix A). *BnaA3.CAT6* and *BnaC7.CAT6* exhibited no expression in most tissues (Appendix A). In order to ascertain the functions of *BnaCATs* in regulating rapeseed against various nutrient stresses, we conducted a transcriptional analysis of their responses under different stress conditions.

The transcriptional identification of the core *CAT* members was significant for the further understanding the functions of *BnaCATs*. The high yield of rapeseed is contingent upon the extensive application of nitrogen fertilizer, yet N use efficiency is low [23]. When N supply is insufficient, plants typically exhibit a suite of adaptive responses to limited N growth conditions [24]. However, the molecular mechanisms underlying the use of N by plants are not fully understood [25]. The transcript levels of *BnaCATs* after low-N treatment were investigated to gain a deeper understanding of their role in assimilating N. Under N stress, the expression of seven *BnaCATs* was found to be significantly altered in the shoots and roots. Six *BnaCATs* (*BnaA4.CAT3*, *BnaAn.CAT4*, *BnaAn.CAT5*, *BnaA9.CAT2a*, *BnaC9.CAT2*, and *BnaA5.CAT4*) exhibited a significant downregulation in both shoots and roots (Figure 7a). In the context of nitrate limitation, the expression of *BnaA4.CAT5* was downregulated in the shoots, while it was induced in the roots (Figure 7a). It can be observed that *BnaA4.CAT5* was the sole gene to exhibit induction in the root under nitrogen deficiency conditions.

Phosphorus (P) is a vital nutrient element for crop growth, occupying a unique and indispensable position in agricultural production [26]. In conditions of phosphate limitation, a total of eight differentially expressed genes (DEGs) were identified in the shoots or roots (Figure 7b). In the shoots, no differential expression of *BnaA3.CAT6* and *BnaC7.CAT6* was observed between sufficient phosphate and insufficient phosphate conditions (Figure 7b). The expression of four *BnaCATs* (*BnaA4.CAT3*, *BnaA4.CAT5*, *BnaC1.CAT4* and *BnaC7.CAT1*) was markedly elevated in the shoots in response to limited phosphate, whereas the expressions of *BnaA5.CAT4* and *BnaC5.CAT4* were significantly repressed (Figure 7b).

No differential expression of *BnaA5.CAT4* was observed in the roots between sufficient and insufficient phosphate conditions (Figure 7b). Under phosphate limitation conditions, the expressions of *BnaA3.CAT6*, *BnaA4.CAT5* and *BnaC7.CAT6* were distinctly downregulated, while *BnaA4.CAT3*, *BnaC5.CAT4* and *BnaC7.CAT1* exhibited higher expression levels (Figure 7b).

Potassium (K) is an important macronutrient in plants [27]. Potassium enhances crop resistance to a variety of biological and abiotic stresses [28,29]. Under the condition of low potassium treatment, the expressions of *BnaC3.CAT6* and *BnaA3.CAT6* were significantly decreased but the expressions of five, especially *BnaA3.CAT1* and *BnaC7.CAT1*, clearly increased in the shoots (Figure 7c). In the roots, K deficiency resulted in an increase in the expressions of *BnaA3.CAT1* and *BnaC7.CAT1*, especially the expression of *BnaC7.CAT1*, significantly increased, while the expression levels of other genes decreased (Figure 7c).

In the shoots, we identified that the DEGs of only *BnaC1.CAT4* and *BnaC3.CAT6* showed higher expression levels under ammonium toxicity than under nitrate sufficiency (Figure 8a), while the DEGs of other *BnaCATs* were significantly downregulated only when ammonium was supplied as the sole N nutrient source (Figure 8a). In the roots, the expressions of most family members (*BnaAn.CAT4*, *BnaA9.CAT2a*, *BnaA4.CAT5*, *BnaA3.CAT1*, *BnaCn.CAT2*, *BnaCn.CAT3*, and *BnaC5.CAT4*) were obviously suppressed only when ammonium was supplied as the sole N nutrient source, whereas the DEGs of *BnaC1.CAT4* and *BnaC3.CAT6* were distinctly upregulated (Figure 8a). It is worth noting that the expression of *BnaC1.CAT4* and *BnaC3.CAT6* was consistently induced in both the shoots and roots (Figure 8a).

Salt stress will induce accumulation of misfolded or unfolded proteins in plants to inhibit their normal growth and development [30]. Salt altered the expression of seven *BnaCATs* in the roots and shoots. Under salt stress, *BnaA3.CAT6* and *BnaC3.CAT6* were obviously downregulated in both roots and shoots (Figure 8b) whereas *BnaAn.CAT4* and *BnaC1.CAT4* were induced in both the shoots and roots (Figure 8b). *BnaA9.CAT8* showed no significant difference in the shoots but it was significantly downregulated in the roots (Figure 8b). *BnaA9.CAT2b* was significantly induced in the shoots but significantly downregulated in the roots (Figure 8b). *BnaA9.CAT2a* showed no significant difference in the roots but was obviously upregulated in the shoots (Figure 8b).

Cadmium (Cd) is known as one of the most hazardous elements in the environment and is a persistent soil constraint toxic to all flora and fauna [31]. To better understand the role of rapeseed CATs in response to cadmium toxicity, we analyzed their transcriptional expression under cadmium toxicity. Under cadmium toxicity, the expression of four *BnaCATs* (*BnaAn.CAT9*, *BnaC1.CAT4*, *BnaC7.CAT1*, and *BnaC9.CAT2*) was elevated in both shoots and roots. Moreover, the expression of *BnaA4.CAT3* was distinctly suppressed in the shoots but induced in the roots (Figure 8c).

## 3. Discussion

*CAT* family members have been reported to play an important role in plant growth and development, nutrient metabolism and stress resistance [32]. *CATs* are widely present in organisms, playing essential roles in regulating plant growth, development and responses to environmental stimuli [33]. In *A. thaliana*, *AtCAT1*, *AtCAT2* and *AtCAT3* control ROS homeostasis by catalyzing H_2_O_2_ decomposition. *AtCAT1* expression is regulated by ABA and MAPK pathways [34], while *AtCAT2* is primarily expressed in leaves and is responsive to light, low temperature and circadian rhythms. *AtCAT3* is highly expressed across developmental stages and is involved in ABA-mediated stomatal regulation [35]. In *Oryza sativa*, *OsCATA* and *OsCATC* are stress responsive; their overexpression enhances drought tolerance, and *OsCATC* phosphorylation by *STRK1* improves both salt and oxidative stress tolerance [36]. Similarly, *CAT* in *Nicotiana tabacum* and *Ipomoea batatas* contribute to H_2_O_2_ homeostasis and stress response [37]. Heterologous expression of *CATs* can further enhance plant stress tolerance; for instance, *wheat CAT* expression in rice increases cold tolerance [38], and *maize CAT2* expression in tobacco enhances pathogen resistance [14]. *CAT* members have also been widely studied in other species. For example, *CATs* had nine members (*CAT1–CAT9*) in *A. thaliana*, and each AAP member only had a single copy. There were totals of 11 *CATs* in rice [39], 19 *CATs* in soybean [40], 12 *CATs* in maize [41], 9 *CATs* in *potato* [42] and 31 *CATs* in wheat [43]. However, the information about Brassicaceae *CATs* is limited so far. In this study, 22 *CATs* were identified in *B. napus*. We used bioinformatics methods to analyze the physical and chemical properties, structure and function of proteins encoded by the *BnaCATs* family. Subsequently, we performed conserved domain, gene structure, gene phylogeny, promoter and synteny analyses. Furthermore, we constructed their protein–protein interaction network. Furthermore, differential expressions of *BnaCATs* under different nutrient conditions were analyzed. These results might provide an integrated insight into the functions of the *CAT* family.

From the analysis on the proteins encoded by the *BnaCAT* family, we found that the subcellular localization of proteins encoded by the *BnaCAT* family is mainly in plasma membranes (Appendix A). All proteins encoded by *BnaCATs* contain 13–15 transmembrane regions, which may be closely related to the regulation of amino acid absorption and transport by the *BnaCATs* (Appendix A). The secondary and three-dimensional structures of the *BnaCAT* family genes showed that the secondary structure of proteins encoded by the *BnaCAT* family genes consists of α -helix, extending chain, β-fold, and random coil elements. These results suggested that the *BnaCATs* may have high similarity in function (Appendix A).

According to phylogenetic analysis, 22 *BnaCATs* were classified into four groups (Figure 1). These results are consistent with previously reported findings in Arabidopsis, tea, and tomato [18,44]. Motifs of *CAT* genes clustered in the same clade were very close, including the number and type (Figure 4). However, different subgroups contained different conserved motifs. Therefore, we speculated that different types of unique motifs may be the main cause of *BnaCATs* functional differentiation.

In the present study, a total of 22 *BnaCATs* (12 on A-subgenome and 10 on C-subgenome) were identified in the rapeseed genome (Table 1). The collinearity results showed that there were a large number of collinearity relationships among *BnaCATs*, and genome-wide replication events/fragment replication may be the main cause of *BnaCATs* gene family amplification. Comparative genomics studies have shown that Brassica species, such as *B. rapa* and *B. oleracea*, experienced triploid events at the genomic level about 20 million years ago [45,46]. As a result, there are three copies of each *A. thaliana* gene in *B. rapa* and *B. oleracea*. Nine *CATs* were identified in the *A. thaliana* genome (Table 1). Theoretically, 27 *CATs* should be included in *B. rapa* and *B. oleracea* after complete genome replication, but only 13 genes were found in *B. oleracea* genome. *B. napus* originates from spontaneous hybridization of the diploid *B. rapa* (ArAr, 2n = 2x = 20) and *B. oleracea* (CoCo, 2n = 2x = 18) [15,16,17]. Therefore, there should be six homologous genes for each Arabidopsis gene in Brassica napus. Unlike conventional theory, only 22 *BnaCATs* were identified in this study, indicating that some *CATs* were lost after whole genome replication. Therefore, we speculated that *CATs* underwent strong selection during the evolution of *B. napus*, and the retained *CATs* should have an important function in *B. napus*. Most of the *AtCATs* in *B. napus* had more than one and less than six direct homologous genes, suggesting that the *CATs* gene family has expanded but contracted during the diversification of *B. napus*. In addition, the homologous genes of *AtCAT7s* were not found in the genome of Brassica napus. The lost *CATs* may be redundant genes that are gradually replaced by other genes with similar functions [47,48]. This research also revealed that all *CATs* in *B. napus* may have undergone rigorous purification selection, which plays a key role in maintaining gene numbers.

Analyzing *cis*-elements in promoter sequences is extremely important for understanding genes regulation and function [21]. This research revealed that the *BnaCAT* family members contained many *cis*-regulatory elements related to phytohormone responsive (ABRE, P-box, TCA-element, TGA-element) (Figure 5). In detail, the content of ABRE is the highest, the number of P-box is lowest (Figure 5). It can be speculated that the *BnaCATs* may be involved in hormone metabolism and regulation. It has been reported that the amino acid transporters can be transported and regulated by hormones [49,50,51]. Furthermore, *cis*-regulatory elements related to stress response (ARE, MBS, LTR, TC-rich repeats, and WUN-motif) can be found in most of *BnaCAT* family members. Some studies indicated that amino acid transporters expression levels were affected by abiotic or biotic factors such as N, salt and drought stress [52,53]. These findings indicated that *CATs* were involved in stress tolerance, especially ABA-mediated biological processes.

We investigated the tissue-specific pattern of *BnaCATs* through BnIR. *BnaA9.CAT8*, *BnaC8.CAT8*, and *BnaAn.CAT9* are widely expressed in most tissues, suggesting that these *BnaCATs* that remained after WGD events are likely functional (Appendix A).

*AtCAT8* in *Arabidopsis* has been reported to provide amino acids to library tissues and participates in early seedling development, and it is widely expressed in all kinds of tissues [7,9]. *BnaA9.CAT8*, *BnaC8.CAT8*, and *BnaAn.CAT9* showed high expression in various tissues, suggesting that they might play important roles in the transfer of amino acids to library tissues (Appendix A). Most of *B. napus* grain nitrogen originates from senescing leaves [53,54,55] and is transported through phloem transport. As a result, phloem loading and transport of amino acids is particularly important for seed yield and quality. It has been reported that *AtAAP8* was mainly expressed in the phloem of the source leaf and located on the plasma membrane of the cell. It is mainly involved in amino acid loading in phloem and nitrogen distribution in source to sink, and then affects the growth of source leaf and seed yield [55]. *PtCAT11* was mainly expressed in phloem and played an important role in amino acid transport between “source” and “sink” tissues during leaf senescence [56]. Our research found that *BnaC7.CAT1*, *BnaA3.CAT1*, *BnaA9.CAT2a*, *BnaCn.CAT2*, *BnaCn.CAT3* and *BnaA5.CAT4* (Appendix A) were highly expressed in senescing leaves. Therefore, we speculated that these genes might be responsible for transport of removable amino acids from source tissues to sink grains for protein synthesis after anthesis.

In order to determine the role of member *BnaCATs* in various stresses, the expression patterns of *BnaCATs* were studied. Seven *BnaCATs* were altered in the upper and lower parts of allotetraploid rapeseed seedlings after N stress. It has been reported in apple that *CATs* were upregulated with the time of nitrogen starvation treatment, and reached the maximum value at 12 h of treatment. And *CAT* was closely related to nitrogen nutrition and may be involved in N metabolism [32]. However, most of the genes in our study were downregulated. We suspected that the amount of gene expression may be closely related to the timing and processing of sampling. In the roots and shoots, many *BnaCATs* were influenced by P stress, while seven *BnaCATs* were differentially expressed in response to K insufficiency.

As the main source of inorganic N in plants, ammonium can promote plant growth at low exogenous levels, but can cause toxicity at high exogenous levels [57]. In this study, we found that the expressions of some *BnaCATs* were altered in response to ammonium toxicity, especially *BnaC3.CAT6*, which indicated *BnaCATs* play an important role in reducing the harm of excessive ammonia to rape plants. The majority of cadmium-responsive *BnaCATs*, except *BnaA4.CAT3*, were induced in both shoots and roots under cadmium toxicity than under the cadmium-free condition. We proposed that the enhanced expression of *BnaCATs* might contribute to efficient amino acid transport and further facilitate the biosynthesis of cadmium chelators [58]. Consequently, it enhances the resistance of plants to cadmium toxicity. Furthermore, the expressions of some genes in shoot increased obviously under salt stress, and might be associated with salt stress.

In addition, this research investigated how some *BnaCATs* were simultaneously responsive to diverse stresses. For example, the expressions of *BnaAn.CAT4* and *BnaC1.CAT4* were affected by four stresses in the shoots or roots simultaneously. Moreover, several *BnaCATs* were involved in three or two stress signals in both roots and shoots of *B. napus*. However, some genes are regulated by a single stress. Thus, our studies suggested that some *BnaCATs* played key roles in responses to various stresses and some were only particular to a specific stress. These all need our further verification.

## 4. Conclusions

Cationic amino acid transporters play key roles in the growth and development of plants. In this study, we identified 22 *BnaCATs*, 13 *BolCATs*, 13 *BraCATs* and 9 *AtCATs* genes via genome-wide analysis. The whole-genome replication or segmental replication was the major driving force of *BnaCATs* evolution. We presented an overview of the chromosomal distributions, gene structures, conserved motifs, *cis*-element, and phylogeny in *BnaCATs*. The transcriptome analysis of *BnaCATs* provides insights for their diverse roles in *Brassica napus* growth and development, especially in nitrate uptake regulation. In summary, our research presented here can serve as useful resources for understanding the structure–function relationship of *BnaCAT* members and reveal regulation mechanisms underlying nutrient stress resistance in the *BnaCAT* family.

## 5. Materials and Methods

### 5.1. Identification of Members of the CAT Gene Family

To identify genome-wide *BnaCATs*, predicted proteins from the *B. napus* genomic database were searched by HMMER v3 using the hidden Markov model (HMM) file that corresponded to the amino acid permease domain (PF13520.8) downloaded from the Pfam database (http://pfam.xfam.org/, accessed on 10 August 2024) as a query [59].The obtained protein sequences had an expected value (E) < 1 × 10^−20^ and contained the amino acid permease domain. Then, we used TBtools to extract our sequence and simplify the ID. Using the amino acid sequences of Arabidopsis *CATs* as source sequences, we built the evolutionary tree with the amino acid sequences of *B. napus* to identify our sequences. In this study, we retrieved the *CAT* sequences using the following databases: The Arabidopsis Information Resource (TAIR10, https://www.arabidopsis.org/, accessed on 10 August 2024) for *A. thaliana* and the Brassica Database (BRAD) v. 4.1 (http://brassicadb.org/brad/, accessed on 14 August 2024) for *B. napus* [60].

### 5.2. Gene Nomenclature of CATs in B. napus

In this study, according to the nomenclature previously reported [61], we renamed *CATs* in *Brassica* species based on the following criterion: genus (one capital letter) + plant species (two lowercase letters) + chromosome (followed by a period) + name of the CAT homologs in *A. thaliana*. For example, *BnaA8.CAT1* represents an *Arabidopsis CAT1* homolog on the chromosome A8 of *B. napus.*

### 5.3. Multiple Sequence Alignment and Phylogeny Analysis of BnaCATs

To analyze the evolutionary relationships, 22 *BnaCATs* and 9 *AtCATs* amino acid residues were aligned via the ClustalW program in MEGA 10.2.2 with default settings. A neighbor-joining tree was then constructed based on the alignment results, and the Interactive Tree of Life (iTOLv5) online tool (https://itol.embl.de/, accessed on 12 August 2024) was finally used to polish the NJ-tree [57].

### 5.4. Molecular Characterization of BnaCATs

The molecular weight (MW, kD), isoelectric point (pI), grand average of hydropathy (GRAVY) and instability index (II) of *BnaCATs* were calculated by the ProtParam tool in ExPASy Server (https://web.expasy.org/protparam/, accessed on 13 August 2024) [58]. To characterize the transmembrane helices of *AtCATs* and *BnaCATs*, we submitted their amino acid sequences to the TMHMM v. 2.0 (http://www.cbs.dtu.dk/services/TMHMM/, accessed on 14 August 2024) program. We employed the online SignalP v. 4.1 (http://www.cbs.dtu.dk/services/SignalP/, accessed on 15 August 2024) [60] to predict the presence of signal peptides in the amino acid sequences of *BnaCATs*. We used the STRING (Search Tool for Recurring Instances of Neighboring Genes) v 11.0 (https://string-db.org, accessed on 14 August 2024) [61] web-server to retrieve and display the repeatedly occurring association networks, including direct (physical) and indirect (function) association, of the CAT proteins in *A. thaliana* and *B. napus*.

### 5.5. Analysis of Evolutionary Selection Pressure and Functional Divergence of BnaCATs

To determine positive or negative (purifying) selection pressures on *BnaCATs*, we calculated the values of Ka, Ks and Ka/Ks. First, we performed pairwise alignment of the *BnaCAT-AtCAT* CDS in an Excel spreadsheet. Then, we submitted the paired CDS sequence to TBtools for the calculation of Ka and Ks [62]. According to Darwin’s evolution theory, Ka/Ks > 1.0 means positive selection, Ka/Ks < 1.0 indicates purifying selection, and Ka/Ks = 1.0 denotes neutral selection. Furthermore, we calculated the divergence time of *BnaCATs* from their progenitors by the following formula: T = Ks/2λ; λ = 1.5 × 10^−8^ for Brassicaceae species [63].

### 5.6. Chromosomal Distribution and Gene Duplication

We used “One Step MCScanX” of TBtools to analyze *CATs* duplication events with genome sequences and gff3 files. The syntenic analysis maps of orthologous *CATs* were constructed using the Dual Systeny Plotter software V2.119 (https://github.com/CJ-Chen/TBtools, accessed on 14 August 2024) [63].

### 5.7. Protein Motif, Gene Structure, and Cis-Element Analyses

The conserved motifs of *BnaCATs* were identified by the MEME Suite web server [63] (v4.12.0, http://meme-suite.org/, accessed on 15 August 2024). The number of motifs was set to 15, and all other parameters were the default ones. The gene structure and conserved domain were visualized via TBtools [59].

The 2000 bp upstream DNA sequence of the 5-UTR of the *BnaCATs* was selected as the promoter sequence. The promoter sequences were uploaded to the PlantCare database (http://bioinformatics.psb.ugent.be/webtools/plantcare/html/, accessed on 15 August 2024) to scan for *cis*-elements. The *cis*-elements from the PlantCare database were subsequently screened manually. The *cis*-elements were visualized via TBtools [59].

### 5.8. Transcriptional Analysis of BnaCATs Under Diverse Nutrient Stresses

We used the BnIR online website (http://yanglab.hzau.edu.cn/BnIR, accessed on 15 August 2024) to analyze the tissue-specific expression of *CATs* in *B. napus*. RNA-seq sequencing libraries were used to sequence via the Illumina HiSeq Xten system (Illumina, San Diego, CA, USA). To identify DEGs (differential expression genes) between two different samples, the expression level of each transcript was calculated according to the transcripts per million reads (TPM) method. RSEM (http://deweylab.github.io/RSEM/, accessed on 1 October 2024) [64] was used to quantify gene abundances. Essentially, differential expression analysis was performed using the DESeq2 [65] or DEGseq [66]. DEGs with |log2FC| ≥ 1 and FDR< 0.05 (DESeq2) or FDR < 0.001 (DEGseq) were considered to be significantly different expressed genes.

### 5.9. Plant Materials and Treatments

The *B. napus* seedlings (ZS11) were germinated in this experiment. Firstly, full-sized oilseed rape seeds were selected and sterilized for 10 min with 1% NaClO, washed with ultrapure water, soaked overnight at 4 °C and sown on seedling trays. After germination, uniform 7-day-old rape seedlings were transplanted into black plastic containers with 10 L of Hochler. The basic nutrition solution contained 1.0 mM KH_2_PO_4_, 5.0 mM KNO_3_, 5.0 mM Ca(NO_3_)_2_·4H_2_O, 2.0 mM MgSO_4_·7H_2_O, 0.050 mM EDTA-Fe, 9.0 μM MnCl_2_·4H_2_O, 0.80 μM ZnSO_4_·7H_2_O, 0.30 μM CuSO_4_·5H_2_O, 0.10 μM Na_2_MoO_4_·2H_2_O and 46 μM H_3_BO_3_.

The rapeseed seedlings were cultivated for 10 days (d) in a chamber under the following conditions: light intensity of 300–320 μmol m^−2^ s^−1^, temperature of 25 °C daytime/22 °C night, light period of 16 h photoperiod/8 h dark and relative humidity of 70%. For the low-nitrate treatment, the 7-day-old uniform *B. napus* seedlings after germination were hydroponically cultivated under high (6.0 mM) nitrate for 10 d, and then were transferred into low (0.30 mM) nitrate solution for 3 d until sampling.

For the ammonium (NH_4_^+^) toxicity treatment, the 7-day-old uniform *B. napus* seedlings after seed germination were hydroponically cultivated under high nitrate (6.0 mM) for 10 d, and then were transferred to N-free conditions for 3 d. Finally, the above seedlings were sampled after exposure to 9.0 mM ammonium for 3 d until sampling.

For the inorganic phosphate (Pi) starvation treatment, the 7-day-old uniform *B. napus* seedlings after seed germination were first hydroponically grown under 250 μM phosphate (KH_2_PO_4_) for 10 d, and then were transferred to 5 μM phosphate for 3 d until sampling.

For the potassium deficiency treatment, the 7-day-old uniform rapeseed seedlings after seed germination were hydroponically cultivated under high (6.0 mM) potassium for 10 d and then were transferred to low (0.05 mM) potassium for 3 d until sampling.

For the salt stress treatment, the 7-day-old uniform *B. napus* seedlings after seed germination were hydroponically cultivated in a NaCl-free solution for 10 d, and were subsequently transferred to 200 mM NaCl for 1 d until sampling.

For the cadmium (Cd) toxicity treatment, the 7-day-old uniform *B. napus* seedlings after seed germination were hydroponically cultivated in a Cd-free solution for 10 d, and then were grown under 10 μM CdCl_2_ for 1 d until sampling.

## Figures and Tables

**Figure 1 ijms-25-12658-f001:**
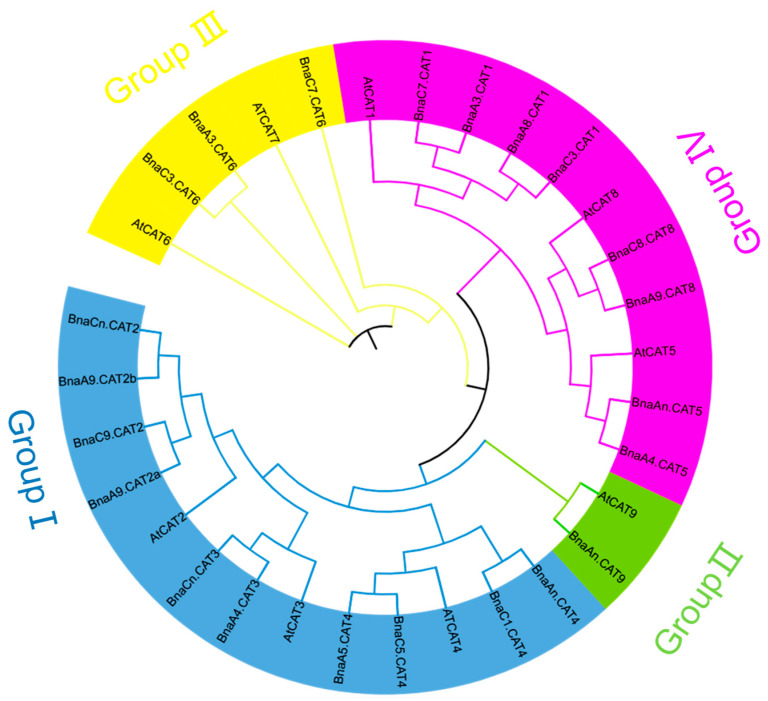
Phylogeny analysis of the *cationic amino acid transporters* (*CATs*) in *Arabidopsis thaliana* and *Brassica napus*. The CAT protein sequences were multi-aligned using the ClustalW program, and then an unrooted phylogenetic tree was constructed using MEGA 10.2.2 with the neighbor-joining method. Overall, 11 *BnaCATs* from *B. napus* and 9 *AtCATs* from *A. thaliana* were clustered into four groups (Group I–IV) based on high bootstrap values signified with different background colors.

**Figure 2 ijms-25-12658-f002:**
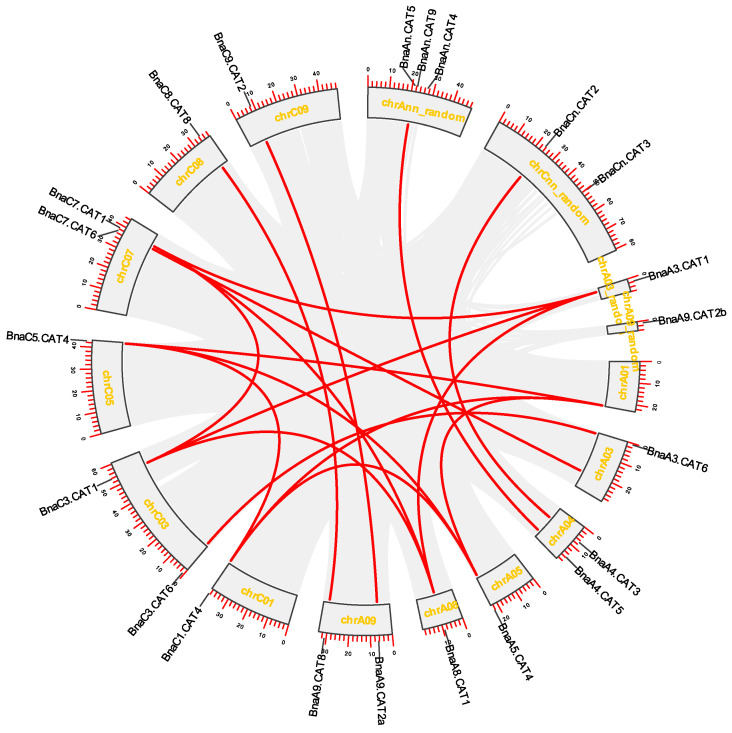
Chromosomal distribution and interchromosomal relationships of *B. napus NLP*. The inner-species collinearity of *BnaCATs*. Gray lines indicate all syntenic blocks in the *B. napus* genome, and the red lines indicate the duplicated *BnaCATs* pairs. The number in the gray box area is the chromosome number.

**Figure 3 ijms-25-12658-f003:**
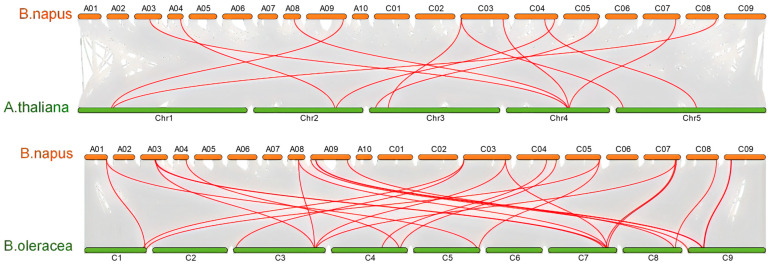
Syntenic *CAT* pairs between *B. napus* and three other plant species, including *A. thaliana*, *B. oleracea*, and *B. rapa.* Gray lines indicate all the collinear blocks in the genome, and the red lines indicate the syntenic *CATs* pairs.

**Figure 4 ijms-25-12658-f004:**
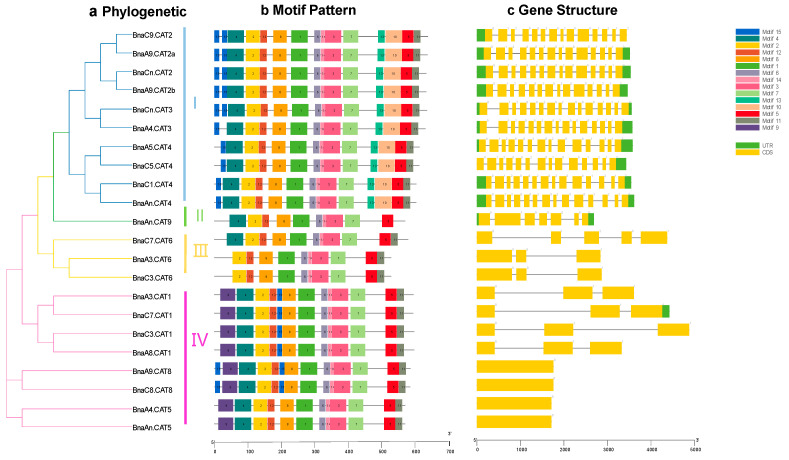
Phylogenetic relationships, architecture of conserved protein motifs and gene structure in *BnaCATs*. (**a**) A phylogenetic tree based on the *BnaCAT* sequences. According to phylogenetic relationships, 22 *BnCATs* were clustered into four groups (I–IV) and are represented with different colors. (**b**) The motif composition of *BnaCATs*. Different colored boxes display different motifs. (**c**) The exon–intron structure of *BnaCATs*. Green boxes indicate UTR regions, yellow boxes indicate exons, blackish-grey lines indicate introns. The bottom scale shows the protein length.

**Figure 5 ijms-25-12658-f005:**
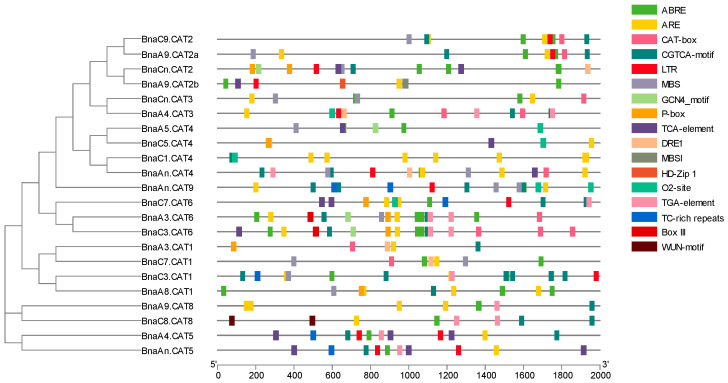
*Cis*-elements that are related to different stress and hormone responses in the putative promoters of *BnaCATs*. *Cis*-elements with similar functions are displayed in the same color. Different color boxes show different identified *cis*-elements.

**Figure 6 ijms-25-12658-f006:**
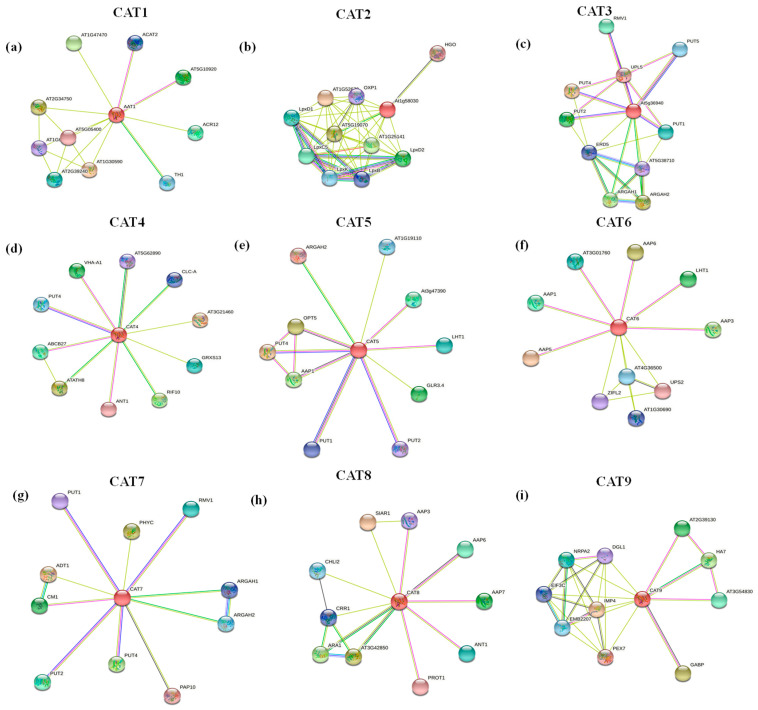
Protein–protein interaction networks of the cationic amino acid transporter (CAT) proteins in *B. napus*. The interaction networks of the CAT1 (**a**), CAT2 (**b**), CAT3 (**c**) CAT4 (**d**), CAT5 (**e**), CAT6 (**f**), CAT7 (**g**), CAT8 (**h**) and CAT9 (**i**) and other proteins were constructed using the STRING web-server.

**Figure 7 ijms-25-12658-f007:**
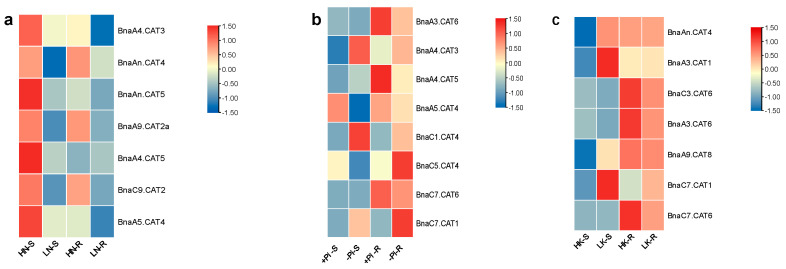
The expression patterns of *BnaCATs* under different N, P and K levels. Expression analysis of *BnaCATs* in the shoots (S) and roots (R) of rapeseed plants grown under different nutrient stresses. Chart legend indicates the values of log2(fold-change). (**a**) HN, high nitrogen (N); LN, low N; (**b**) +Pi, add phosphorus; −Pi, lack phosphorus; (**c**) HK, high potassium; LK, low K.

**Figure 8 ijms-25-12658-f008:**
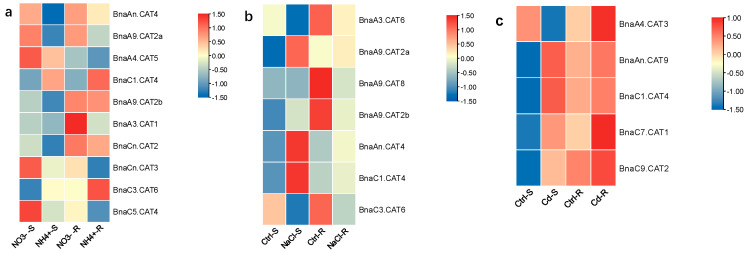
The expression patterns of *BnaCATs* under different nitrogen regimes, salt and Cd toxicity. (**a**) Expression analysis of *BnaCATs* in shoot and root under nitrate sufficiency (NO^3−^) and ammonium toxicity (NH_4_^+^). (**b**) Expression analysis of *BnaCATs* in root and shoot under salt toxicity (NaCl) and salt-free (Ctrl) conditions. (**c**) Expression analysis of *BnaCATs* in root and shoot under Cd toxicity (Cd) and Cd-free (Ctrl) conditions.

**Table 1 ijms-25-12658-t001:** Copy number of the Cationic amino acid transporters (CATs) in Arabidopsis and three Brassica species.

Gene Name	*Arabidopsis thaliana* (125 Mb)	*Brassica rapa* (465 Mb)	*Brassica oleracea* (485 Mb)	*Brassica napus* (1130 Mb)
*CAT1*	1	2	2	4
*CAT2*	1	2	2	4
*CAT3*	1	1	1	2
*CAT4*	1	2	2	4
*CAT5*	1	1	1	2
*CAT6*	1	2	2	3
*CAT7*	1	1	1	0
*CAT8*	1	1	1	2
*CAT9*	1	1	1	1
Total	9	13	13	22

**Table 2 ijms-25-12658-t002:** Molecular characterization of the cationic amino acid transporters (*CATs*) in *Arabidopsis thaliana* and *Brassica napus*.

Gene ID	Gene Name	Block	Amino Acids (aa)	CDS (bp)	Ka	Ks	Ka/Ks	Divergent Times (Mya)
*BnaC07g36580D*	*BnaC7.CAT1*	U	595	1788	0.065	0.49	0.13	16.56
*BnaC03g64380D*	*BnaC3.CAT1*	U	598	1797	0.076	0.50	0.15	16.91
*BnaA03g58530D*	*BnaA3.CAT1*	U	596	1791	0.063	0.48	0.13	16.07
*AT1G58030.1*	*AtCAT2*	D	635	1908				
*BnaA09g12110D*	*BnaA9.CAT2a*	D	638	1917	0.056	0.46	0.12	15.37
*BnaA09g53040D*	*BnaA9.CAT2b*	D	634	1905	0.056	0.43	0.12	14.46
*BnaCnng25140D*	*BnaCn.CAT2*	D	634	1905	0.056	0.43	0.13	14.42
*BnaC09g12080D*	*BnaC9.CAT2*	D	639	1920	0.057	0.43	0.13	14.64
*AT5G36940.1*	*AtCAT3*	S	609	1830				
*BnaCnng50730D*	*BnaCn.CAT3*	S	636	1911	0.064	0.36	0.17	12.07
*BnaA04g07800D*	*BnaA4.CAT3*	S	632	1899	0.067	0.36	0.18	12.05
*AT3G03720.2*	*AtCAT4*	F	600	1803				
*BnaC01g40540D*	*BnaC1.CAT4*	F	604	1815	0.046	0.34	0.13	11.51
*BnaA05g32770D*	*BnaA5.CAT4*	F	614	1845	0.046	0.33	0.14	11.12
*BnaC05g48070D*	*BnaC5.CAT4*	F	613	1842	0.046	0.33	0.14	11.02
*BnaAnng23630D*	*BnaAn.CAT4*	F	604	1815	0.042	0.34	0.12	11.52
*AT2G34960.1*	*AtCAT5*	J	569	1710				
*BnaAnng19530D*	*BnaAn.CAT5*	J	570	1713	0.045	0.55	0.08	18.61
*BnaA04g20450D*	*BnaA4.CAT5*	J	571	1716	0.046	0.55	0.08	18.35
*AT5G04770.1*	*AtCAT6*	R	583	1752				
*BnaA03g01440D*	*BnaA3.CAT6*	R	529	1590	0.036	0.59	0.061	19.86
*BnaC03g01740D*	*BnaC3.CAT6*	R	529	1590	0.036	0.53	0.067	17.98
*BnaC07g34120D*	*BnaC7.CAT6*	R	580	1743	0.42	2.14	0.19	71.58
*AT1G17120.1*	*AtCAT8*	A	590	1773				
*BnaC08g37970D*	*BnaC8.CAT8*	A	586	1761	0.041	0.75	0.054	25.29
*BnaA09g45150D*	*BnaA9.CAT8*	A	586	1761	0.037	0.68	0.054	22.92
*AT1G05940.1*	*AtCAT9*	A	569	1710				
*BnaAnng20490D*	*BnaAn.CAT9*	A	571	1716	0.037	0.27	0.13	9.23

## Data Availability

All the data and plant materials in relation to this work can be obtained through contacting with the corresponding author Dr. Ying-Peng Hua (yingpenghua@zzu.edu.cn).

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
