# Peer review of "Genome-Wide Identification of the CAT Genes and Molecular Characterization of Their Transcriptional Responses to Various Nutrient Stresses in Allotetraploid Rapeseed"

_ijms, 2024, doi:10.3390/ijms252312658_

Round 1
Reviewer 1 Report
Comments and Suggestions for Authors
The paper contains very little data and only performs genome-wide analysis, so I would recommend rejecting it. Furthermore, the paper lacks subcellular localization figures and other interaction studies such as Y1H and Y2H. Given the lack of a functional study to supplement the data and IJMS's reputation, I would suggest rejecting this paper. However, I think there are many other MDPI journals where the paper can be submitted.
Comments on the Quality of English LanguageThe language can be improved from its current level by taking help of any native English speaker.
Author Response
General comment: The paper contains very little data and only performs genome-wide analysis, so I would recommend rejecting it. Furthermore, the paper lacks subcellular localization figures and other interaction studies such as Y1H and Y2H. Given the lack of a functional study to supplement the data and IJMS's reputation, I would suggest rejecting this paper. However, I think there are many other MDPI journals where the paper can be submitted.
Response: We are appreciated very much for your warm work and the constructive comments on our manuscript. We studied your comments carefully and have made corrections which we hope meet with approval. This study performed a systematic genome-wide analysis and molecular characterization of the CAT gene family members in Brassica napus. Brassica napus is an important oil crop in China and has a great demand for nitrogen. Cationic amino acid transporters (CAT) play a key role in amino acid absorption and transport in plants. However, the CAT family genes have not been reported in Brassica napus so far. In this study, genome-wide analysis identified 22 CAT members in Brassica napus genome. Based on phylogenetic and synteny analysis, BnaCATs were classified into four groups (Group I-Group IV). The members in the same subgroups showed that similar physiochemical characteristics, intron/exon and motif patterns. By evaluating cis-elements in the promoters, we identified some cis-elements related to hormone, stress and plant development. Darwin’s evolutionary analysis indicated that BnaCATs might have experienced strong purifying selection pressure. The BnaCAT gene family in Brassica napus may have undergone gene expansion, the chromosomal location of BnaCATs indicated that whole genome replication or segmental replication may play a major driving role. Differential expression pattern of BnaCATs under nitrate limitation, phosphate shortage, potassium shortage, cadmium toxicity, ammonium excess, and salt stress conditions indicated that they were responsive to different nutrient stresses. In summary, these findings provide a comprehensive survey of the BnaCAT gene family and lay a foundation for the further functional analysis of family members.

Reviewer 2 Report
Comments and Suggestions for Authors
The manuscript titled “Genome-wide identification of the cationic amino acid transporter (CAT) genes and molecular characterization of their transcriptional responses to various nutrient stresses in allotetraploid rapeseed” is a comprehensive study focusing on an essential aspect of Brassica napus, an important oil crop. By identifying and analyzing CAT gene family members, the study provides valuable insights into nutrient transport and stress response mechanisms, which can contribute to improving crop yield and resilience. The results have significant implications for enhancing nitrogen use efficiency (NUE) and adapting B. napus to challenging environmental conditions. After addressing the minor revisions suggested, the manuscript could be consider for publication.

A thorough proofread for minor grammatical errors and typos is recommended.
Author Response
General comment: The manuscript titled “Genome-wide identification of the cationic amino acid transporter (CAT) genes and molecular characterization of their transcriptional responses to various nutrient stresses in allotetraploid rapeseed” is a comprehensive study focusing on an essential aspect of Brassica napus, an important oil crop. By identifying and analyzing CAT gene family members, the study provides valuable insights into nutrient transport and stress response mechanisms, which can contribute to improving crop yield and resilience. The results have significant implications for enhancing nitrogen use efficiency (NUE) and adapting B. napus to challenging environmental conditions. After addressing the revisions suggested, I strongly recommend this manuscript for publication.
Response: We appreciate for your warm work and thanks very much for your positive and constructive comments on our manuscript. Your comments are considerably valuable and very helpful for revising and improving our paper, particularly on the experimental design for proceeding the project in the future. We have studied your comments carefully and have made corrections with tracked changes which we hope meet with your approval. The point-by-point responses to your comments are also listed as be.
Comment 1: While the manuscript provides a solid background on the role of CAT genes in amino acid transport and nutrient stress responses, there is limited discussion on previous studies that investigated similar genes in other crops. Including more references to studies on CAT genes in Arabidopsis, wheat, and soybean would provide a more comprehensive understanding of how B. napus compares to other species.
Response: Thank you very much for your kind suggestion.
According to you request, we have added CAT genes for wheat, soybean, and cucumber.
In wheat, ten CAT proteins were identified,the study showed that TaCAT proteins may be related to the response to various stresses, are cytoplasm localized, and may function as antioxidant enzymes[56]. Four proteins have been identified in soybean,play an important role in plant defense ,development and senescence[57]. Four candidate CsCAT genes were identified in cucumber.Based on the expression pattern comparison, CsCATs exhibited an expression pattern similar to Arabidopsis counterparts[58]. In addition, the ectopic expression of maize CAT2 (ZmCAT2) in tobacco can induce CAT activity, improving pathogen resistance[59]
References
Zhang Y, Zheng L, Yun L, Ji L, Li G, Ji M, Shi Y, Zheng X. Catalase (CAT) Gene Family in Wheat (Triticum aestivum L.): Evolution, Expression Pattern and Function Analysis. Int J Mol Sci. 2022 Jan 4;23(1):542.
Aleem M, Aleem S, Sharif I, Aleem M, Shahzad R, Khan MI, Batool A, Sarwar G, Farooq J, Iqbal A, Jan BL, Kaushik P, Feng X, Bhat JA, Ahmad P. Whole-Genome Identification of APX and CAT Gene Families in Cultivated and Wild Soybeans and Their Regulatory Function in Plant Development and Stress Response. Antioxidants (Basel). 2022 Aug 22;11(8):1626.
Hu L, Yang Y, Jiang L, Liu S. The catalase gene family in cucumber: genome-wide identification and organization. Genet Mol Biol. 2016 Jul-Sep;39(3):408-15. doi: 10.1590/1678-4685-GMB-2015-0192. Epub 2016 Jul 25.
Polidoros AN, Mylona PV, Scandalios JG. Transgenic tobacco plants expressing the maize Cat2 gene have altered catalase levels that affect plant-pathogen interactions and resistance to oxidative stress. Transgenic Res. 2001 Dec;10(6):555-69.
Comment 2:The manuscript presents detailed bioinformatic analyses and experimental data on the transcriptional responses of CAT genes. However, the figures and tables, while informative, could benefit from additional clarification.
Response: Thank you very much for your request.
According to your advice, we have modified the content in the table and figures to make them clearer。
Comment 3: Clarification of Gene Expression Analysis: The expression analysis of CAT genes under nutrient stress conditions is comprehensive, but the presentation of results, especially in the figures, can be overwhelming. The heatmaps and bar charts, although informative, could benefit from a more concise presentation.
Response: Thank you very much for your request。
According to your advice, we have modified the figures in the results section to make them clearer and clarified the heatmaps and bar charts.
Comment 4:Suggestion: Comparative Analysis with Other Species: The manuscript providesan extensive analysis of CAT genes in B, napus, but lacks a detailed comparison with similar findings in other model plants like Arabidopsis thaliana or agriculturally important species such as wheat or maize. This would help contextualize the findings and highlight species-specific or conserved mechanismsin CAT-mediated nutrient stress responses.
Response: Thank you very much for your request.
According to your advice, we have added content to the Discussion.
CAT are widely present in organisms, playing essential roles in regulating plant growth, development, and responses to environmental stimuli [60]. In Arabidopsis thaliana, AtCAT1, AtCAT2, and AtCAT3 control ROS homeostasis by catalyzing H2O2 decomposition. AtCAT1 expression is regulated by ABA and MAPK pathways[61], while AtCAT2 is primarily expressed in leaves and is responsive to light, low temperature, and circadian rhythms. AtCAT3 is highly expressed across developmental stages and is involved in ABA-mediated stomatal regulation [62]. In Oryza sativa, OsCATA and OsCATC are stress-responsive; their overexpression enhances drought tolerance, and OsCATC phosphorylation by STRK1 improves both salt and oxidative stress tolerance [63]. Similarly, CAT in Nicotiana tabacum and Ipomoea batatas contribute to H2O2 homeostasis and stress response [64]. Heterologous expression of CAT can further enhance plant stress tolerance; for instance, wheat CAT expression in rice increases cold tolerance[65], and maize CAT2 expression in tobacco enhances pathogen resistance [66].
Reference
Zhou, Y.-B.; Liu, C.; Tang, D.-Y.; Yan, L.; Wang, D.; Yang, Y.-Z.; Gui, J.-S.; Zhao, X.-Y.; Li, L.-G.; Tang, X.-D.; et al. The Receptor-Like Cytoplasmic Kinase STRK1 Phosphorylates and Activates CatC, Thereby Regulating H2O2 Homeostasis and Improving Salt Tolerance in Rice. Plant Cell 2018, 30, 1100–1118.
Du, Y.-Y.; Wang, P.; Chen, J.; Song, C.-P. Comprehensive Functional Analysis of the Catalase Gene Family in Arabidopsis thaliana. J. Integr. Plant Biol. 2008, 50, 1318–1326.
Su, T.; Wang, P.P.; Li, H.J.; Zhao, Y.W.; Lu, Y.; Dai, P.; Ren, T.Q.; Wang, X.F.; Li, X.Z.; Shao, Q.; et al. The Arabidopsis catalase triple mutant reveals important roles of catalases and peroxisome-derived signaling in plant development. J. Integr. Plant Biol. 2018, 60, 591–607.
Schmidt, R.; Mieulet, D.; Hubberten, H.-M.; Obata, T.; Hoefgen, R.; Fernie, A.R.; Fisahn, J.; Segundo, B.S.; Guiderdoni, E.; Schippers, J.H.M.; et al. SALT-RESPONSIVE ERF1 Regulates Reactive Oxygen Species-Dependent Signaling during the Initial Response to Salt Stress in Rice. Plant Cell 2013, 25, 2115–2131.
Willekens, H.; Villarroel, R.; Van Montagu, M.; Inzé, D.; Van Camp, W. Molecular identification of catalases from Nicotianaplumbaginifolia (L.). FEBS Lett. 1994, 352, 79–83.
Matsumura, T.; Tabayashi, N.; Kamagata, Y.; Souma, C.; Saruyama, H. Wheat catalase expressed in transgenic rice can improve tolerance against low temperature stress. Physiol. Plant. 2002, 116, 317–327.
Polidoros, A.; Mylona, P.; Scandalios, J. Transgenic tobacco plants expressing the maize Cat2 gene have altered catalase levels that affect plant-pathogen interactions and resistance to oxidative stress. Transgenic Res. 2001, 10, 555–569.
Comment 5:Some figures, particularly Figure 3, contain light-colored text and excessive blank spaces, which reduces readability. Darken the text and reduce blank spaces to improve clarity and visual appeal.
Response: Thank you very much.
According to your advice, we have corrected the text and reduce blank spaces in the revised manuscript, which is also as follows:
Comment 6:The nomenclature for B. napus genes should be consistently applied throughout the text and figures. Ensure that all gene names, such as BnaCAT1, are consistently formatted in both the text and figures for clarity.
Response: Thank you very much for your kind advice.
According to your advice, we have corrected the B.napus name .
According to the protein sequences of the AtCATs gene family in Arabidopsis, 22 BnaCAT members have been identified in the B.napus genome. According to the se-quence on the B.napus chromosome , the 22 BnaCAT genes are renamed,such as BnaC9.CAT2, BnaC3.CAT6
Comment 7:In some sections, such as Line 311, the citation format is inconsistent. Standardize citation formatting to align with the journal’s guidelines.
Response: Thank you very much for your kind suggestion. According to your advice, we have corrected citation formatting.
The conserved motifs in A. thaliana and B.napus were extracted by the MEME program based on protein sequences. In total, 15 conserved motifs were identified, among the 15 conserved domains that we defined, we found that the amino acid sequences of the motifs 1,2,3,5,6,7,8,12 and 14 had the highest identity among all the BnaCATs (Figure 4), and thus might be used as indicators of the CAT family members. The genetic classification, revelated that the motif composition of CAT proteins within the four groups exhibited some similarities, while there were also some differences among the groups (Figure 4). For instance, motif 15 was specific to group I, while motif 9 was specific to group IV (Figure 4). This indicated that the CAT sequence is evolutionarily conserved but differentiated.
Comment 8:Phrases like "The data" (Line 217) are vague and could confuse readers. Clarify what specific data is being referred to in such instances.
Response: Thank you very much for your kind suggestion.
According to your advice, we have corrected the phrases.
To evaluate the sequence diversity of BnaCATs, the exon–intron structures of each BnaCAT were detected. In detail, the number of introns and exons varies among each group of BnaCAT (Figure 4). It was observed that similar structures were typically found within the same group (Figure 4). The number of introns in Group I is 13, with the exception of BnaA4.CAT3 (Figure 4). The BnaAn.CAT9 sequence exhibited six introns (Figure 4). The Group III genes exhibited two or four introns (Figure 4). The number of introns present in the Group IV genes exhibited a range from zero to two (Figure 4). These results indicated the clusters of BnaCATs had a similar intron/exon pattern. Studies on the conserved motif composition, gene structure and phylogenetic relationship have demonstrated that BnaCAT proteins have very conserved amino acid residues, and members within the group may have similar functions.

Reviewer 3 Report
Comments and Suggestions for Authors
The manuscript by Xiao-qian Du et al. presents results obtained after in silico conducted research on identification the genome-wide CAT genes in B. napus, genomic properties and transcriptional responses of CAT gene members to N stresses and the transcriptional responses of CATs to other nutrient stresses, including phosphate limitation, boron deficiency, cadmium toxicity, and salt stress.
I have reviewed this manuscript and I think it is worth to be publish although some points should be improved. In my opinion the evidence provided by the authors is not sufficient to meet their conclusions.
Specific comments and questions to be addressed:
First of all, some methological details should be included, for example:
In chapter Materials and Methods only one B. napus cultivars (Zhongshuang 11) was used. Why? Was it selected according to NUE index value or some other trait?
Please write more precisely how the transcriptional analysis of BnaAAPs was done.
In Discussion chapter the Authors indicated that four stresses in the shoots or roots at the same time were observed but in methods there is no mentioned at all about the procedure of stress induction on seedlings. Please clarify it in the methods.
It is difficult to evaluate the correctness of the results, because there is no precisely described methodology.
Line 62 – correct the words “in B.napusm”
Line 68 – put the commas within the sentences
The text should be reedited and write more concise in English.
Author Response
Point-to-point response to Reviewer #3
General comment: The manuscript by Xiao-qian Du et al. presents results obtained after in silico conducted research on identification the genome-wide CAT genes in B. napus, genomic properties and transcriptional responses of CAT gene members to N stresses and the transcriptional responses of CATs to other nutrient stresses, including phosphate limitation, boron deficiency, cadmium toxicity, and salt stress.
I have reviewed this manuscript and I think it is worth to be publish although some points should be improved. In my opinion the evidence provided by the authors is not sufficient to meet their conclusions.
Response: We appreciate for your warm work and thanks very much for your positive and constructive comments on our manuscript. Your comments are considerably valuable and very helpful for revising and improving our paper, particularly on the experimental design for proceeding the project in the future. We have studied your comments carefully and have made corrections in red with tracked changes which we hope meet with your approval. The point-by-point responses to your comments are also listed as below.
Comment 1:In chapter Materials and Methods only one B. napus cultivars (Zhongshuang 11) was used. Why? Was it selected according to NUE index value or some other trait?
Response: Thank you very much for your request.
Oil Crops Research Institute of Chinese Academy of Agricultural Sciences (Wuhan, China) develops ZS11, which is a widely cultivated semi-winter rapeseed variety and glucosinolate content (Sun et al., 2017). ZS11 shows
strong tolerance against diverse nutrient stresses, such as boron deficiency (Hua et al. 2016) and cadmium toxicity(Zhang et al. 2019).
Zhou T, Wu PJ, Chen JF, Du XQ, Feng YN, Hua YP. Pectin demethylation-mediated cell wall Na+ retention positively regulates salt stress tolerance in oilseed rape. Theor Appl Genet. 2024 Feb 21;137(3):54. doi: 10.1007/s00122-024-04560-w. PMID: 38381205.
Hua YP, Wu PJ, Zhang TY, Song HL, Zhang YF, Chen JF, Yue CP, Huang JY, Sun T, Zhou T. Genome-Scale Investigation of GARP Family Genes Reveals Their Pivotal Roles in Nutrient Stress Resistance in Allotetraploid Rapeseed. Int J Mol Sci. 2022 Nov 21;23(22):14484. doi: 10.3390/ijms232214484. PMID: 36430962; PMCID: PMC9698747.
Sun F, Fan G, Hu Q et al (2017) The high-quality genome of Bras sica napus cultivar “ZS11” reveals the introgression history in semi-winter morphotype. Plant J 92:452–468
Comment 2:Please write more precisely how the transcriptional analysis of BnaAAPs was done.
Response: Thank you very much for your kind suggestion.
In view of this valuable suggestion, we have made content changes of the structure and content of the transcriptional analysis in the revised manuscript.
Comment 3:In Discussion chapter the Authors indicated that four stresses in the shoots or roots at the same time were observed but in methods there is no mentioned at all about the procedure of stress induction on seedlings. Please clarify it in the methods.
Response: Thank you very much for your request.
According to your advice, we have added the four stresses in the shoots or roots at the same time in the Materials and methods section, which is also as follows:
Plant Materials and Treatments
The B. napus seedlings (ZS11) germinated in this experiment. Firstly, full sized oilseed rape seeds were selected and sterilised for 10 minutes with 1% NaClO, washed with ultrapure water, soaked overnight at 4 °C and sown on seedling trays. After germination, uniform 7-day-old rape seedlings were transplanted into black plastic containers with 10 litres of Hochler. The basic nutrition solution contained 1.0 mM KH2PO4, 5.0 mM KNO3, 5.0 mM Ca(NO3)2·4H2O, 2.0 mM MgSO4·7H2O, 0.050 mM EDTA-Fe, 9.0 μM MnCl2·4H2O, 0.80 μM ZnSO4·7H2O, 0.30 μM CuSO4·5H2O, 0.10 μM Na2MoO4·2H2O, and 46 μM H3BO3.
The rapeseed seedlings were cultivated for 10 days (d) in a chamber under the following conditions: light intensity of 300–320 μmol m− 2 s− 1, temperature of 25°C daytime/22°C night, light period of 16 h photoperiod/ 8 h dark, and relative humidity of 70%. For the low nitrate treatment, the 7-d- old uniform B. napus seedlings after germination were hydroponically cultivated under high (6.0 mM) nitrate for 10 d, and then were transferred into low (0.30 mM) nitrate solution for 3 d.
For the ammonium (NH4+) toxicity treatment, the 7-d-old uniform B. napus seedlings after seed germination were hydroponically cultivated under high nitrate (6.0 mM) for 10 d, and then were transferred to N-free condition for 3 d. Finally, the above seedlings were sampled after exposure to 9.0 mM ammonium for 3 d.
For the inorganic phosphate (Pi) starvation treatment, the 7-d-old uniform B. napus seedlings after seed germination were first hydroponically grown under 250 μM phosphate (KH2PO4) for 10 d, and then were transferred to 5 μM phosphate for 3 d.
For the potassium deficiency treatment, the 7-d-old uniform rapeseed seedlings after seed germination were hydroponically cultivated under high (6.0 mM) potassium for 10 d and then were transferred to low (0.05 mM) potassium for 3 d.
For the salt stress treatment, the7-d-old uniform B. napus seedlings after seed germination were hydroponically cultivated in a NaCl-free solution for 10 d, subsequently were transferred to 200 mM NaCl for 1 d.
For the cadmium (Cd) toxicity treatment, the 7-d-old uniform B. napus seedlings after seed germination were hydroponically cultivated in a Cd-free solution for 10 d, and then were grown under10 μM CdCl2 for 1d.
Comment 4:It is difficult to evaluate the correctness of the results, because there is no precisely described methodology.
Response: Thank you very much for your request.
In view of this valuable suggestion, we have readjusted the structure and content of the Results in the revised manuscript.
Comment 5:Line 62 – correct the words “in B.napusm”
Response: Thank you very much for your question.
According to your advice, we have corrected in B.napusm into in B.napus in the revised manuscript.
However, there are few systematic analyses of CATs in B. napus. Therefore, it is of great importance to CATs members in B.napus to analyse the nutritional physiological and biological characteristics of rapeseed.
Comment 6:Line 68 – put the commas within the sentences.
Response: Thank you very much for your good advice.
According to your advice, we have put the commas within the sentences.
Bioinformatics and molecular biology methods were used to identify compare and analyse the expression of the B. napus CAT (named BnaCAT). Through the statistics of transmembrane region of BnaCAT proteins, active site prediction, phylogenetic analysis, protein interaction and expression pattern exploration, it can provide partial reference for the related studies of amino acid transporter and nitrogen nutrient metabolism in rapeseed.
Comment 7:The text should be reedited and write more concise in English.
Response: Thank you very much for your good advice.
Thank you very much for your kind suggestion. We have tried our best to polish the language in the revised manuscript. These changes will not influence the content and framework of this paper. And here we did not list the changes but marked in red in the revised manuscript.
To further clarify the potential functions of CATs in B. napus, MEME was used to identify 15 conserved motifs. We found that the amino acid sequences of the motifs 1,2,3,5,6,7,8,12 and 14 had the highest identity among all the BnaCATs (Figure 4), and thus might be used as indicators of the CAT family members. The genetic classification revelated that four groups CAT proteins exhibited similarities, while there were also differences among the groups (Figure 4). For instance, motif 15 was specific to group I and motif 9 was specific to group IV (Figure 4). This indicated that the CAT sequence is evolu-tionarily conserved but differentiated.
To evaluate the sequence diversity of BnaCATs, the exon–intron structures of each BnaCAT were detected. In detail, the number of introns and exons varies among each group of BnaCAT (Figure 4). It was observed that similar structures were typically found within the same group (Figure 4). The number of introns in Group I is 13, with the exception of BnaA4.CAT3 (Figure 4). The BnaAn.CAT9 sequence exhibited six introns (Figure 4). The Group III genes exhibited two or four introns (Figure 4). The number of introns present in the Group IV genes exhibited a range from zero to two (Figure 4). These results indicated the clusters of BnaCATs had a similar intron/exon pattern. Studies on the conserved motif composition, gene structure and phylogenetic relationship have demonstrated that BnaCAT proteins have very conserved amino acid residues, and members within the group may have similar functions.
These results indicated the clusters of BnaCATs had a similar intron/exon pattern.
To explore the selective pressure on BnaCATs, the non-synonymous/synonymous mutation ratio (Ka/Ks) was calculated; Ka/Ks > 1.0 indicates positive selection, Ka/Ks = 1.0 indicates neutral selection, and Ka/Ks < 1.0 indicates purifying selection(Table 2).

Round 2
Reviewer 1 Report
Comments and Suggestions for Authors
Could be accepted after this rigorous revision.